# Spectrum-to-Kernel Translation for Accurate Blind Image Super-Resolution

**Guangpin Tao**[1*]  **Xiaozhong Ji**[1,2]  **Wenzhuo Wang**[1]  **Shuo Chen**[3]  **Chuming Lin**[2]

**Yun Cao**[2]  **Tong Lu**[1†]  **Donghao Luo**[2]  **Ying Tai**[2]

[1]National Key Lab for Novel Software Technology, Nanjing University
[2]Tencent Youtu Lab [3]RIKEN Center for Advanced Intelligence Project

## Abstract

Deep-learning based Super-Resolution (SR) methods have exhibited promising performance under non-blind setting where blur kernel is known. However, blur kernels of Low-Resolution (LR) images in different practical applications are usually unknown. It may lead to significant performance drop when degradation process of training images deviates from that of real images. In this paper, we propose a novel blind SR framework to super-resolve LR images degraded by arbitrary blur kernel with accurate kernel estimation in frequency domain. To our best knowledge, this is the first deep learning method which conducts blur kernel estimation in frequency domain. Specifically, we first demonstrate that feature representation in frequency domain is more conducive for blur kernel reconstruction than in spatial domain. Next, we present a Spectrum-to-Kernel (S2K) network to estimate general blur kernels in diverse forms. We use a Conditional GAN (CGAN) combined with SR-oriented optimization target to learn the end-to-end translation from degraded images' spectra to unknown kernels. Extensive experiments on both synthetic and real-world images demonstrate that our proposed method sufficiently reduces blur kernel estimation error, thus enables the off-the-shelf non-blind SR methods to work under blind setting effectively, and achieves superior performance over state-of-the-art blind SR methods, averagely by **1.39dB**, **0.48dB** on commom blind SR setting (with Gaussian kernels) for scales $2\times$ and $4\times$, respectively.

## 1 Introduction

Single Image Super-Resolution (SISR) is a low-level visual task to recover High-Resolution (HR) images from its degraded Low-Resolution (LR) counterparts. Deep learning methods have significantly promoted SR research and achieved remarkable results on benchmarks [16, 26, 27, 13, 29, 35, 17]. Most of these works are based on the degradation model assuming that the LR image is down-sampled with blur kernel (*e.g.*, bicubic kernel) and additional noise from its HR source. Recent studies [6, 11] show that pre-trained SR model is sensitive to the degradation of LR image. The mismatch for Gaussian kernels may cause SR result either over-sharpening or over-smoothing, and for motion kernels, it may bring unpleasant jitters and artifacts. Blind SR is to super-resolve a single LR image with its kernel unknown. It has attracted more and more attention due to its close relationship with the real scene. However, compared with the advanced non-blind SR researches, existing blind SR methods are insufficient to meet the needs of various degradation processes in real general scenes.

---

[*]Work done during an intership at Tencent Youtu Lab.
[†] Corresponding author (lutong@nju.edu.cn).

35th Conference on Neural Information Processing Systems (NeurIPS 2021).

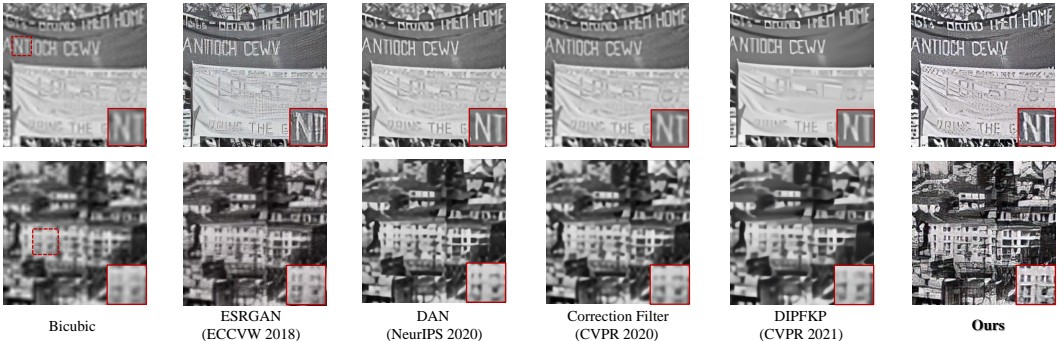

| Bicubic | ESRGAN (ECCVW 2018) | DAN (NeurIPS 2020) | Correction Filter (CVPR 2020) | DIPFKP (CVPR 2021) | **Ours** |

Figure 1: 4× blind SR results comparison on real historic images. Our method achieves the best visual perception in recovery of real-scene objects such as clearer text shape and building structures. LR patches are cropped from images in http://vllab.ucmerced.edu/wlai24/LapSRN/.

Some representative works have been proposed on this challenging problem. Bell-Kligler *et al.* estimate the kernel with internal-Generative Adversarial Network (GAN) on the basis of maximum similarity of patches across scales, whose performance is limited even with handcraft constraints for the weak prior fails to capture the kernel-related features directly [2]. Gu *et al.* propose an Iterative Kernel Correction (IKC) framework to correct the kernel estimation from the mismatch between kernel and SR result in an iterative way [6]. IKC is based on Maximum A Posteriori (MAP), in which the estimator depends on the posterior of the pre-trained restorer. As demonstrated in [14], due to the accuracy limitation of the restorer, the maximum performance of the estimator cannot be guaranteed. Ji *et al.* introduce the frequency consistency constraint between the generated down-sampled image and the real input to estimate the blur kernel in spatial domain [11]. However, its training requires a certain amount of data with the same degradation and fails to capture blur diversities in various directions, limiting its application for a specific input LR image degraded with an arbitrary kernel.

In this paper, we capture the convolution degradation feature of the local distribution in LR images from a novel perspective. Based on convolution theorem and sparsity analysis, we demonstrate the Fourier frequency spectrum of degraded LR image provides a robust shape structure of the degradation kernel in the frequency domain, which can be utilized to reconstruct the unknown kernel more accurately and robustly. Based on the above analysis, we design a Spectrum-to-Kernel (S2K) translation network to directly obtain the kernel interpolation result from degraded LR's frequency spectrum. As far as we know, this is the first work to estimate kernel thoroughly in the frequency domain instead of in the spatial domain previously. The experimental results demonstrate that our method accurately estimates the target kernel and has a high universality capacity for diverse kernels. Moreover, when combined with the existing non-blind methods, we achieve the best results in quality evaluation and visual perception on synthetic and real images against existing blind SR methods. Figure 1 compares 4× blind SR results with recently proposed methods on real historical images.

In general, our main contributions are summarized as follows:

- We theoretically demonstrate that for common sparse kernels, it is more conducive to reconstruct it in the frequency domain instead of the commonly used spatial domain.
- We analyze the impacts of kernel estimation optimization objectives for blind SR performance and design a novel network S2K to estimate kernel from Fourier amplitude spectrum.
- When plugged into existing non-blind SR models, our proposed pipeline achieves the best performance on both synthetic and real-world images against existing blind SR methods.

## 2 Related Work

**Non-blind SR** Under the non-blind setting, HR image $\mathbf{I}_{HR}$ is recovered from paired given paired LR image $\mathbf{I}_{LR}$ and kernel $\mathbf{k}$. Many successful pieces of research have been proposed where the bicubic kernel is used to construct paired training data, so they belong to the non-blind setting in essence. We divide those kernel-aware SR methods into two categories. (1) Multiple Degradation SR (MDSR) based methods concatenate degraded image $\mathbf{I}_{LR}$ with stretched blur kernel $\mathbf{k}$ as SR model's input [33, 31] or conduct intermediate spatial feature transform according to the corresponding kernel

to get target SR result [6, 31]. (2) Kernel Modeling SR (KMSR) based methods incorporate the blur kernel in the model's training parameters by constructing the paired kernel modeling dataset for the known test kernels to get a specific SR model for test data's degradation restoration [36, 10]. Inconsistency between $\mathbf{I}_{LR}$ and blur kernel $\mathbf{k}$ leads to significant performance deterioration in test phase [33]. We propose a spectrum-to-kernel framework (S2K) that provides accurate kernel estimation to help both MDSR and KMSR based non-blind methods work well under blind settings.

**Blind SR**   Under blind setting, $\mathbf{I}_{HR}$ is recovered from $\mathbf{I}_{LR}$ degraded with unknown kernel. Since it is more closely related to real application scenarios, blind SR has attracted research attention for a long time [21], mainly involving transforming unknown degraded image into target $\mathbf{I}_{HR}$'s bicubic down-sampled one, and non-blind SR methods with kernel estimation from spatial input $\mathbf{I}_{LR}$.

In the former category, Hussein *et al.* modify the $\mathbf{I}_{LR}$ with unknown $\mathbf{k}$ into that down-sampled from $\mathbf{I}_{HR}$ by bicubic kernel [9]. Maeda *et al.* train a correction network to get the corresponding clean LR and construct pseudo pair to train the SR model [19]. In the latter category, Bell-Kligler *et al.* utilize the recurrence of patches across scales to generate the same-degraded down-sampled image as the test input [2]. However, the estimation accuracy is limited even with many handcrafted constraints due to the weak prior. Gu *et al.* use a kernel prediction network with a correction network taking the current kernel prediction and SR result of the pre-trained SR model as input to correct the kernel estimation iteratively [6]. Huang *et al.* adopt an alternating optimization algorithm with two modules restorer and estimator [8]. Ji *et al.* obtain the approximation of the blur kernel by utilizing the frequency domain consistency to estimate kernel in an unsupervised way [11]. Liang *et al.* propose Flow-based Kernel Prior (FKP) learning an invertible mapping between the kernel distribution and a tractable distribution to generate reasonable kernel initialization, improving the optimization stability [15].

Different from previous kernel estimation methods in the spatial domain, we demonstrate Fourier spectrum in *frequency domain* can provide a more effective kernel shape structure, which contributes to accurate and robust estimation for arbitrary kernels. Further, utilizing the shape structure relationship of the sparse kernel between the frequency domain and the spatial domain, our method can estimate diverse blur kernels end-to-end with implicit cross-domain translation. Besides, we propose SR-oriented kernel estimation optimization objection and obtain a more accurate estimation of arbitrary unknown blur kernels directly and independently. When combining S2K with existing advanced non-blind SR methods, we achieve the best quantitative and qualitative performance on synthetic and real-world images against existing blind SR methods.

## 3   Method

In this section, we first formulate the problem, then theoretically analyze the advantage of blur kernel estimation from the frequency domain and the feasibility of end-to-end translation learning. Then we introduce the pipeline and component of the proposed S2K network in detail. Finally, we present the blind SR pipeline combined with the proposed kernel estimation module.

### 3.1   Problem Formulation

Same as [33, 34, 11], we assume the image degradation process from $\mathbf{I}_{HR}$ to $\mathbf{I}_{LR}$ is as follows:

$$\mathbf{I}_{LR} = (\mathbf{I}_{HR}\downarrow_s) \otimes \mathbf{k} + \mathbf{n}, \tag{1}$$

where $\mathbf{k}$ denotes degradation kernel, and $\mathbf{n}$ is additional noise[3]. $\downarrow_s$ and $\otimes$ are down-sampling and convolution operations. Under non-blind setting, both $\mathbf{k}$ and $\mathbf{n}$ are known, denoting the pre-trained SR model as $\mathbf{R}$, the upper bound of SR result $\mathbf{I}^*_{SR}$ can be expressed as:

$$\mathbf{I}^*_{SR} = \mathbf{R}(\mathbf{I}_{LR} \mid \mathbf{k}, \mathbf{n}). \tag{2}$$

Under blind setting, the degradation prior needs to be obtained from $\mathbf{I}_D$. In this paper, we mainly focus on the estimation of degradation kernel from the degraded input $\mathbf{I}_{LR}$, because the extra denoising method can be easily plugged in from the previous denoising methods [7, 10]. Theoretically, the optimal kernel estimation target $\mathbf{k}^*$ for blind SR performance is defined as follows:

$$\mathbf{k}^* = \arg\min_{\mathbf{k}_{est}} \|\mathbf{R}(\mathbf{I}_{LR} \mid \mathbf{k}_{est}) - \mathbf{I}^*_{SR}\|. \tag{3}$$

---

[3]Above variables are all matrices, and blur kernel size is far less than LR image size.

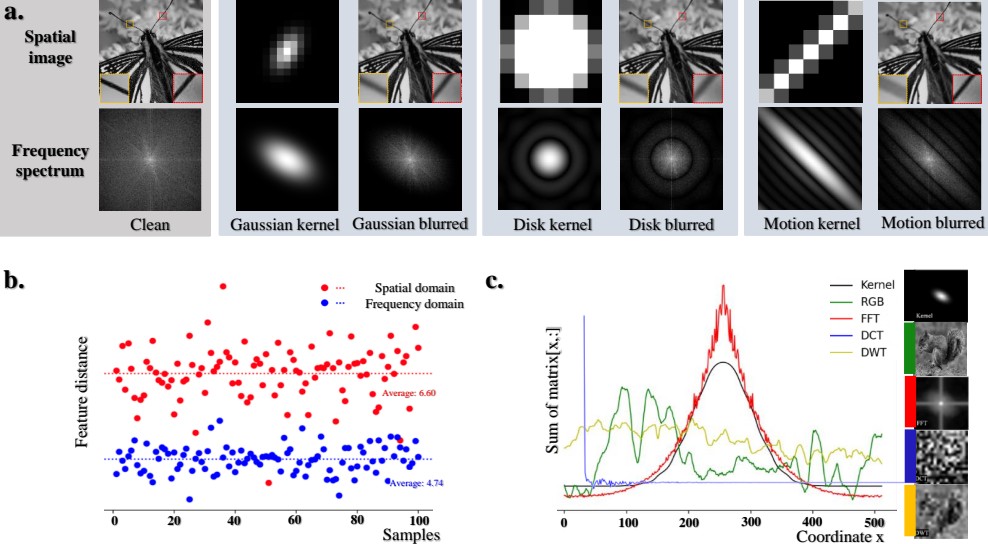

Figure 2: (a). Relationship between $\mathbf{I}_{HR}$, $\mathbf{k}$ and $\mathbf{I}_{LR}$ in frequency and spatial domain. Fourier spectrum of $\mathbf{I}_{LR}$ indicates shape of kernel in frequency domain, which provides a significant and robust kernel representation. However, spatial distribution show minor connection between $\mathbf{I}_{LR}$ and $\mathbf{k}$, and the blurring caused by the same kernel $\mathbf{k}$ in different positions of the $\mathbf{I}_{LR}$ may be different (as indicated in red and yellow boxes), leading to the instability of blur kernel estimation training. (b). Pre-trained VGG19 feature distance between kernel estimation's input and output in spatial domain and frequency domain respectively. (c). For a single image, the pixel intensity sum (along one dimension) distance between estimation input and the target kernel in different domains.

Accurate kernel estimation is the key to make the most of the powerful performance of existing non-blind methods under the blind setting. However, it is difficult for its ill-posed property. Meanwhile, the non-blind SR models are very sensitive to the consistency of input $\{\mathbf{I}_{LR}, \mathbf{k}_{est}\}$ pairs, and subtle degradation inconsistencies may cause obvious artifacts [6]. We explored the influence of the kernel estimation optimization target on the performance of subsequent non-blind SR to ensure the consistency of $\mathbf{k}_{est}$ and $\mathbf{I}_D$, and get the result close to the upper bound $\mathbf{I}_{SR}^*$ in the following parts.

### 3.2 Spectrum-to-Kernel

Different from previous kernel estimation methods in spatial domain, S2K is the first deep-learning-based attempt to conduct kernel estimation entirely in the frequency domain. Shape structure is conducive to the reconstruction task learning: Chen *et al.* obtain better face SR results with shape prior [5], Ma *et al.* maintain the gradients structure, which proves to be beneficial to the SR result [18]. Some researchers utilize structure information to improve the performance of image inpainting [25, 22]. We find the kernel representation obtained from the Fourier spectrum can provide more robust and favorable feature from shape structure than spatial patches. We below give a formal proof that kernel estimation in the frequency domain is more conducive than in the spatial domain under the reasonable assumption that blur kernels of different forms satisfy sparsity.

**Advantages of the Frequency Domain**  In frequency domain (Fourier frequency domain unless specified), denoting clean images, degraded images, and degradation kernels as $\mathbf{S}$, $\mathbf{G}$ and $\mathbf{F}$ respectively. According to the convolution theorem, denoting hadamard product as $\odot$, we have:

$$\mathbf{G} = \mathbf{F} \odot \mathbf{S}. \tag{4}$$

We adopt $\phi(\mathbf{X}, \mathbf{Y})$ to measure shape similarity of two matrix variables $\mathbf{X}, \mathbf{Y}$ [12, 32] as follows:

$$\phi(\mathbf{X}, \mathbf{Y}) = \|\delta_\tau(\mathbf{X} - \mathbf{Y})\|_0, \tag{5}$$

where $\delta_\tau(\cdot)$ is a truncated function to ignore the extremely small divergence in each elements and $\|\cdot\|_0$ denotes $\ell_0$-norm operator. When the absolute value of element $\mathbf{x}$ in $(\mathbf{X} - \mathbf{Y})$ is less than $\tau$

(positive but very small), it returns 0, otherwise returns the absolute value of $\mathbf{x}$. Under the assumption that common kernels are sparse in both the frequency and spatial domains, we compare the shape structure similarity of the input and output for kernel estimation in the frequency and spatial domains, respectively. In the frequency domain, we have:

$$
\begin{aligned}
\phi(\mathbf{G}, \mathbf{F}) &= \sum_{\mathbf{F}_{ij} < \tau} \mathbf{sign}(\delta_\tau(\mathbf{F}_{ij}\mathbf{S}_{ij} - \mathbf{F}_{ij})) + \sum_{\mathbf{F}_{ij} \geq \tau} \mathbf{sign}(\delta_\tau(\mathbf{F}_{ij}\mathbf{S}_{ij} - \mathbf{F}_{ij})) \\
&\leq \sum_{\mathbf{F}_{ij} < \tau} \mathbf{sign}(\delta_\tau(\mathbf{F}_{ij}(\mathbf{S}_{ij} - 1))) + \sum_{\mathbf{F}_{ij} \geq \tau} \mathbf{sign}(\delta_\tau(\mathbf{F}_{ij})) \\
&= 0 + \sum_{\mathbf{F}_{ij} \geq \tau} \mathbf{sign}(\delta_\tau(\mathbf{F}_{ij})) = \|\delta_\tau(\mathbf{F})\|_0.
\end{aligned}
\tag{6}
$$

Here, if $x$ is positive, $\mathbf{sign}(\mathbf{x}) > 0$, else if $x$ is negative, $\mathbf{sign}(\mathbf{x}) < 0$, else $\mathbf{sign}(\mathbf{x}) = 0$, and $i, j$ denote position coordinates in spectrum. Eq. 6 gives the upper bound of the distance of the shape structure from the degraded input to the output kernel in the frequency domain. While in the spatial domain, denoting the zero-padded degradation kernel with $\mathbf{k}_p$ which has the same size of the degraded image $\mathbf{I}_{LR}$, we have:

$$
\begin{aligned}
\phi(\mathbf{k}_p, \mathbf{I}_{LR}) &= \sum_{[\mathbf{k}_p]_{ij} < \tau} \mathbf{sign}(\delta_\tau([\mathbf{k}_p]_{ij} - [\mathbf{I}_{LR}]_{ij})) + \sum_{[\mathbf{k}s_p]_{ij} > \tau} \mathbf{sign}(\delta_\tau([\mathbf{k}_p]_{ij} - [\mathbf{I}_{LR}]_{ij})) \\
&\geq \sum_{[\mathbf{k}_p]_{ij} < \tau} \mathbf{sign}(\delta_\tau([\mathbf{k}_p]_{ij} - [\mathbf{I}_{LR}]_{ij})) \geq \sum_{\substack{[\mathbf{k}_p]_{ij} < \tau, \\ [\mathbf{I}_{LR}]_{ij} > 2\tau}} \mathbf{sign}(\delta_\tau([\mathbf{k}_p]_{ij} - [\mathbf{I}_{LR}]_{ij})) \\
&\geq \|\delta_{2\tau}(\mathbf{I}_{LR})\|_0 - \|\delta_\tau(\mathbf{k}_p)\|_0,
\end{aligned}
\tag{7}
$$

which gives the lower bound of the distance of the shape structure on the spatial domain. Further, since both $\mathbf{F}$ and $\mathbf{k}_p$ are sparse, $\|\delta_\tau(\mathbf{F})\|_0$ and $\|\delta_\tau(\mathbf{k}_p)\|_0$ are significantly smaller than $\|\delta_{2\tau}(\mathbf{I}_{LR})\|_0$ due to the non-sparsity of $\mathbf{I}_{LR}$. In this case, $\|\delta_\tau(\mathbf{F})\|_0 \ll \|\delta_{2\tau}(\mathbf{I}_{LR})\|_0 - \|\delta_\tau(\mathbf{k}_p)\|_0$. Thus we have:

$$
\phi(\mathbf{G}, \mathbf{F}) \ll \phi(\mathbf{k}_p, \mathbf{I}_{LR}).
\tag{8}
$$

Eq. 8 reveals that the divergence between the blur kernel and its corresponding degraded image in the frequency domain is significantly smaller than that in the spatial domain. As illustrated in Figure 2, compared with the spatial domain, $\mathbf{G}$ and $\mathbf{F}$ have a more significant and robust kernel shape correlation, which is conducive for the kernel reconstruction in the frequency domain.

After demonstrating the shape advantage for general kernel estimation from the frequency domain over the spatial domain, we further analyze the cross-domain relationship of the kernel's shape structure. Taking the Gaussian kernel as the example, according to Fourier transform property of 2D Gaussian function, kernel in the frequency domain is also in Gaussian function form, and its variances are inversely proportional with that in spatial. Using the notation in Figure 3 left, we have:

$$
\sigma_u = \mathbf{C}_1 / \sigma_x,
\tag{9}
$$

$$
\sigma_v = \mathbf{C}_2 / \sigma_y.
\tag{10}
$$

Here, $\mathbf{C}_1, \mathbf{C}_2$ refer to two constants[4]. It reveals that it is feasible to learn the cross-domain translation implicitly and estimate spatial kernel from the degraded image in the frequency domain with spectrum in an end-to-end manner. Experimentally, we find it still works for motion and disk kernels.

### 3.3 Spectrum based Kernel Estimation

Based on the analysis above, we propose a new kernel estimation method from the frequency domain, which makes full use of the shape structure from $\mathbf{I}_{LR}$'s spectrum to reconstruct the kernel . We further utilize kernel's shape relationship to learn the cross-domain translation, thus estimating the unknown spatial kernel from the frequency spectrum end-to-end.

---

[4]More details provided in supplementary materials.

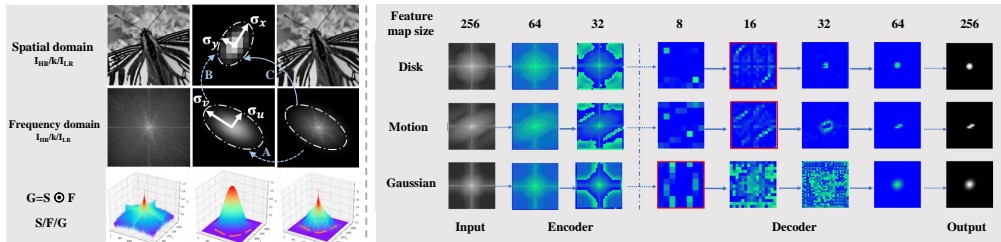

Figure 3: **Lelf**: **A.** Shape structure from **G** is conducive to the reconstruction of kernel spectrum **F** in the frequency domain. **B.** Shape correlation of the Gaussian kernel between Fourier domain and spatial domain: variances are inversely proportional, respectively. **C.** Based on **A** and **B**, it is feasible to estimate the spatial kernel **k** from the degraded image's spectrum **G**. **Right:** Visualization of intermediate feature maps of different sizes of the pre-trained S2K. We show the shape structure from the degraded LR image (as marked in red) are conducive to kernel estimation in frequency domain.

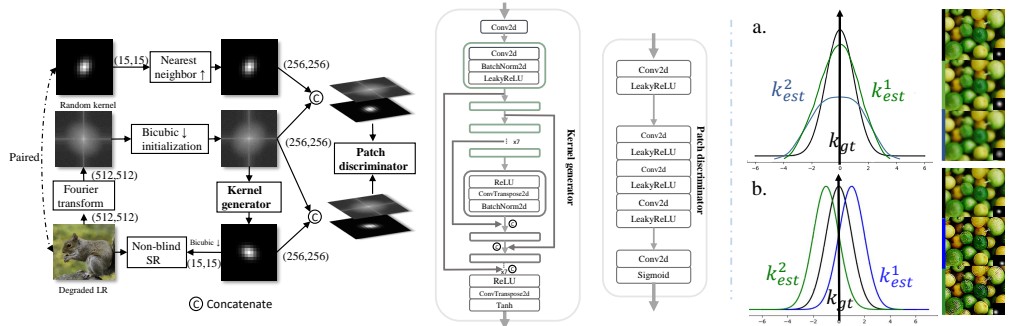

Figure 4: Left: Pipeline of proposed S2K with a conditional kernel generator and a patch discriminator. Right: Illustration of two toy examples to show the limitations of using pixel loss $\ell_1$ (a.) and relative shape loss $\ell_s$ (b.) alone respectively, where optimization objective of kernel estimation is inconsistent with subsequent SR performance. In case (a.), $\mathbf{k}_{est}^1$ and $\mathbf{k}_{est}^2$ share the same $\ell_1$ distance to $\mathbf{k}_{gt}$, but obtain more blurry SR result with $\mathbf{k}_{est}^2$ than $\mathbf{k}_{est}^1$. In case (b.), $\mathbf{k}_{est}^1$ and $\mathbf{k}_{est}^2$ have the same shape distance to $\mathbf{k}_{gt}$, but inconsistent with SR results. We find the combination of the pixel-level value loss and relative shape loss is more beneficial to subsequent SR performance with estimated kernel.

**Kernel Generator**    Unlike the previous methods estimating kernel's parameter from spatial input, we obtain the target kernel in spatial from the frequency amplitude. The Generator is an encoder-decoder structure (Figure 4) and takes the $256 \times 256$ single-channel amplitude spectrum of the LR image as the input. After through an input layer with stride 2, the feature is sent into a U-net where a mirror connection is between each of the seven down-sampling Conv layers and up-sampling transposed Conv layers of the same feature size. Finally, we obtain the single-channel estimation kernel map through the output layer. As discussed in Figure 4 right, we use relative shape loss $\ell_s$ and smoothing loss $\ell_{reg}$ in addition to the $\ell_1$ loss to ensure that the estimated kernel close to ground truth and *favorable* for SR results. For input pair $\{\mathbf{s}_i, \mathbf{k}_i\}$ ($\mathbf{s}_i$ denotes spectrum of $\mathbf{I}_{LRi}$), to make the generator output $G(\mathbf{s}_i)$ closer to target $\mathbf{k}_i$, the generator's loss $\ell_G$ consists of three parts:

$$\ell_G = \mathbb{E}_{\mathbf{s}_i, \mathbf{k}_i} [\underbrace{\lambda_1 \|\mathbf{k}_i - G(\mathbf{s}_i)\|_1}_{\ell_1} + \underbrace{\lambda_2 [D(\mathbf{s}_i; G(\mathbf{s}_i)) - 1]^2}_{\ell_s} + \underbrace{\lambda_3 \mathcal{R}(G(\mathbf{s}_i))}_{\ell_{reg}}], \tag{11}$$

where $( ; )$ denotes feature concatenating and $D$ denotes the discriminator, $\mathcal{R}$ denotes regularization item, $\lambda_i$ denotes weight of respective loss component. The first loss $\ell_1$ pushes $G(s_i)$ pixel distance closer to target, the second loss $\ell_s$ makes $G(s_i)$ satisfy the target kernel's spatial surface shape. We use adversarial loss to make the generated results conform to target kernel's distribution, serving as the shape loss. The third total variation loss $\ell_{reg}$ regularizes the output with smoothing regularization.

**Patch Discriminator**    We use a fully-convolutional patch discriminator to learn the distribution of different kernel forms. To make the training more stable, we adopt variant of LSGAN [20] where the least square loss is calculated between the patch embeddings of the predicted kernel and the

| Method | Kernel | DIV2K[28] | | | Flicker2K[1] | | |
|---|---|---|---|---|---|---|---|
| | | 2× | 3× | 4× | 2× | 3× | 4× |
| RCAN finetuned | Gaussian | 24.92 / 0.6903 | 23.39 / 0.6304 | 22.37 / 0.5977 | 24.88 / 0.6832 | 23.46 / 0.6333 | 22.47 / 0.6013 |
| ZSSR | | 24.90 / 0.6900 | 23.36 / 0.6302 | 22.36 / 0.5974 | 24.91 / 0.6907 | 23.43 / 0.6332 | 22.46 / 0.6013 |
| DeblurGAN w. RCAN | | 25.38 / 0.7090 | 23.87 / 0.6463 | 22.64 / 0.6039 | 25.39 / 0.7091 | 23.85 / 0.6465 | 22.72 / 0.6080 |
| KernelGAN | | 25.53 / 0.7130 | 23.57 / 0.6370 | 22.97 / 0.6130 | 25.48 / 0.7126 | 23.66 / 0.6398 | 23.09 / 0.6174 |
| FCA w. RCAN | | 28.29 / 0.8160 | 26.19 / 0.7382 | 24.93 / 0.6873 | 28.45 / 0.8160 | 26.30 / 0.7414 | 24.81 / 0.6905 |
| FKP | | 28.13 / 0.8023 | 26.05 / 0.7219 | 24.82 / 0.6748 | 28.40 / 0.8188 | 26.22 / 0.7412 | 24.68 / 0.6912 |
| IKC | | 28.45 / 0.8366 | 26.84 / 0.7595 | 25.56 / 0.7100 | 28.46 / 0.8421 | 26.72 / 0.7589 | 25.64 / 0.7146 |
| **S2K w. SFTMD** | | 29.31 / 0.8359 | 27.19 / 0.7590 | 25.84 / 0.7096 | 29.38 / 0.8411 | 26.96 / 0.7582 | 25.79 / 0.7143 |
| **S2K w. RCAN** | | 29.77 / 0.8444 | 27.47 / 0.7674 | 26.05 / 0.7139 | 29.91 / 0.8469 | 27.39 / 0.7639 | 26.10 / 0.7145 |
| RCAN finetuned | Motion | 23.34 / 0.6522 | 22.06 / 0.6080 | 22.04 / 0.5782 | 23.43 / 0.6535 | 22.11 / 0.5992 | 21.49 / 0.5803 |
| DeblurGAN w. RCAN | | 23.30 / 0.6555 | 22.24 / 0.6096 | 21.24 / 0.5778 | 23.22 / 0.6540 | 22.10 / 0.6018 | 21.40 / 0.5801 |
| ZSSR | | 23.33 / 0.6506 | 22.25 / 0.6074 | 21.32 / 0.5785 | 24.91 / 0.6907 | 23.43 / 0.6332 | 22.46 / 0.6013 |
| KernelGAN | | 22.40 / 0.6080 | 22.17 / 0.6035 | 20.45 / 0.5414 | 22.40 / 0.6115 | 22.00 / 0.5935 | 20.56 / 0.5363 |
| IKC | | 24.96 / 0.7543 | 22.27 / 0.6357 | 22.11 / 0.6330 | 24.70 / 0.7352 | 21.95 / 0.6244 | 21.89 / 0.6284 |
| Pan *et al.* [23] w. SFTMD | | 24.10 / 0.6729 | 20.04 / 0.5539 | 19.21 / 0.5318 | 21.04 / 0.5996 | 19.90 / 0.5529 | 18.80 / 0.5168 |
| Pan *et al.* [24] w. SFTMD | | 21.50 / 0.5673 | 20.37 / 0.5113 | 19.63 / 0.4765 | 21.28 / 0.5238 | 20.22 / 0.5182 | 19.83 / 0.4932 |
| FKP | | 24.38 / 0.6897 | 22.39 / 0.6273 | 21.02 / 0.5887 | 24.43 / 0.6902 | 22.41 / 0.6268 | 21.10 / 0.7011 |
| **S2K w. SFTMD** | | 30.95 / 0.8870 | 28.28 / 0.8118 | 26.95 / 0.7593 | 31.21 / 0.8971 | 28.32 / 0.8212 | 26.70 / 0.7557 |

Table 1: Quantitative [PSNR↑ / SSIM↑] comparison results of fidelity-oriented SR model for $2\times$, $3\times$, $4\times$ up-sampling. The best performance is shown in red and the second best in blue.

corresponding ground truth (both concat. with $\mathbf{s}_i$). Discriminator's training loss can be written as:

$$\ell_D = \mathbb{E}_{\mathbf{s}_i, \mathbf{k}_i}[(D(\mathbf{s}_i; G(\mathbf{s}_i)))^2 + (D(\mathbf{s}_i; \mathbf{k}_i) - 1)^2]. \tag{12}$$

### 3.4 Pipeline of Blind SR

Based on the proposed S2K kernel estimation module as shown in Figure 4. We further introduce our basic pipeline for blind SR. For kernels defined by continuous parameters *i.e.*, Gaussian kernels and disk kernels (3 and 1 parameters), both MDSR and KMSR [36] (without kernel expansion) based non-blind models are applicable. For motion kernels with more discrete and irregular form, we use MDSR based SR model [6] to get blind SR result taking $\mathbf{I}_{LR}$ and estimated kernel $\mathbf{k}$ as input.

## 4 Experiments

**Experiment Setup** In theory, our method can handle general sparse kernels. We select the two most commonly used blur kernels in image processing, *i.e.*, Gaussian and motion kernels to conduct blind SR experiments. We set Gaussian kernel size as 15 and variance between 1.0 and 3.0, rotation angle between 0 and $2\pi$. We randomly select from 500 motion kernels using code from [3], and the kernel size is 23, exposure time is set in [0.15, 0.3], anxiety is 0.005. In addition, we add estimation of disk kernels and use Matlab function *fspecial* to randomly generate disk kernel according to the input radius between 1.0 and 3.0. All kernel-aware methods compared are trained on the *same* dataset degraded with random kernels (three types of random kernels are estimated separately). We use DIV2K [28] training set to train S2K and the non-blind SR model. Then, we use random blurred DIV2K validation set, 100 images in Flicker2K [1] as test datasets in the synthetic experiment, and conduct $2\times$, $3\times$, $4\times$ SR comparison and kernel estimation accuracy comparison. Finally, we use our pre-trained estimation model of Gaussian kernels to perform $4\times$ SR test on the real image from RealSR [4] and historical photos. We provide qualitative and quantitative comparisons with state-of-the-art methods. We set the weights of loss item in generator as $\lambda_1 = 100$, $\lambda_2 = 1$, $\lambda_3 = 1$ respectively. For optimization, we use Adam with $\beta_1 = 0.5$, $\beta_2 = 0.999$, and learning rate is 0.001.

### 4.1 Blind SR on Synthetic Images

We combine proposed S2K with non-blind methods and provide comparisons with existing methods in Table 1. Due to the inconsistency of training (clean dataset) and test (unknown degradation) domain, the performance of ordinary ideal kernel-awareless SR models (*e.g.*, RCAN, ZSSR) suffers a significant deterioration on the test data. The performance of blind methods trying to correct test LR image to get clear LR image (*e.g.*, DeblurGAN w. RCAN) is limited because down-sampling in degradation process makes the recovery of high-frequency components more difficult in LR image. KernelGAN and FKP obtain more accurate predictions than bicubic interpolation kernel, but the

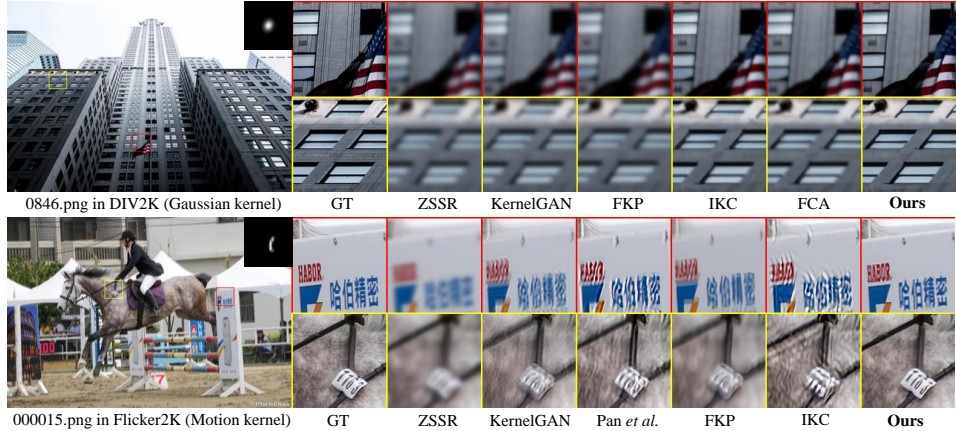

| | 0846.png in DIV2K (Gaussian kernel) | GT | ZSSR | KernelGAN | FKP | IKC | FCA | **Ours** |

| | 000015.png in Flicker2K (Motion kernel) | GT | ZSSR | KernelGAN | Pan *et al.* | FKP | IKC | **Ours** |

Figure 5: $2\times$ blind SR results on synthetic images with state-of-the-art methods. For *Gaussian* and *motion* blurred images, our method obtains results with *more clarity* and *less jitter effects*.

| Methods | Gaussian kernel $|K_{Est} - K_{Gt}|$ | | | | | Motion kernel $|K_{Est} - K_{Gt}|$ | | | | | Disk kernel $|K_{Est} - K_{Gt}|$ | | | | | $\sum \mathcal{D}_v(K_{Est}, K_{Gt})$ Gaussian/Motion/Disk | $\sum \mathcal{D}_s(K_{Est}, K_{Gt})$ Gaussian/Motion/Disk |
|---|---|---|---|---|---|---|---|---|---|---|---|---|---|---|---|---|---|
| $K_{Gt}$ | | | | | | | | | | | | | | | | – | – |
| Xu *et al.* | | | | | | | | | | | | | | | | 0.2813/0.4555/0.7159 | 144.2/429.09/335.5 |
| Pan *et al.* | | | | | | | | | | | | | | | | 0.1209/0.3705/0.1696 | 16.82/446.33/115.4 |
| KernelGAN | | | | | | | | | | | | | | | | 0.2146/0.3922/0.3310 | 27.27/515.76/297.2 |
| FKP | | | | | | | | | | | | | | | | 0.3839/0.4851/0.5229 | 129.0/699.47/577.8 |
| S2K | | | | | | | | | | | | | | | | **0.0274/0.0672/0.0089** | **0.6788/22.27/0.6237** |

Figure 6: Kernel estimation comparison on $3\times$ degraded LR of DIV2K dataset with random kernels. Compared with [30, 23, 2, 15], Ours can get more accurate blur kernel estimation results close to GT.

inadequate estimation accuracy degrades the subsequent SR model's performance. Using the same pre-trained SFTMD model, we can obtain better results against IKC, for our estimation kernel model is not based on the pre-trained MDSR model but in the frequency domain. FCA is unable to deal with the kernel estimation of the specific degraded image while our method can estimate more general kernels more accurately without these restrictions. Visual comparison results are provided in Figure 5.

**Kernel Estimation**  We provide the $\ell_1$ distance $\mathcal{D}_v$ and relative shape distance $\mathcal{D}_s$ as in Eq. 13 between predict kernel and the ground truth on 100 test $I_{LR}$ images. Here $h$, $w$ denote the height and width of the kernel, $i$, $j$ denote position indices, respectively.

$$\mathcal{D}_v(X, Y) = \frac{1}{h*w} \sum_{i,j} |X_{i,j} - Y_{i,j}|, \; \mathcal{D}_s(X, Y) = \sum_{i,j} X_{i,j} \log(\frac{X_{i,j}}{Y_{i,j}}). \tag{13}$$

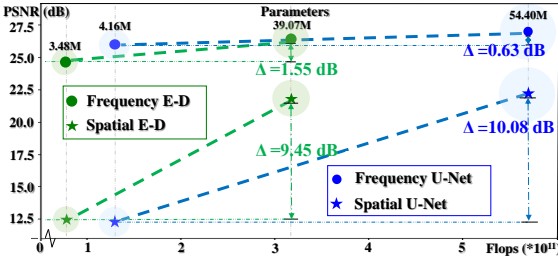

Figure 7: SR performance with different generator models. Here, E-D denotes U-net removing the mirror skip connection. We use Unet-8-64, Unet-5-32 and ED-8-64, ED-5-32 (*i.e.*, arch.-downsample/residual blocks-channels). Estimation in frequency domain is less affected by model capability, and the PSNR drop from model reduction is far less than that in spatial domain, showing the advantages of frequency domain estimation.

The results are shown in Figure 6. Compared with existing single image kernel estimation methods, our proposed method can more accurately estimate Gaussian, motion, and disk kernels qualitatively and quantitatively. Further, to verify the advantage and generality of estimation in the frequency domain, we use different reconstruction model architectures and sizes, respectively, and compare the

Table 2: Ablation study on loss components.

| Settings | | | Metrics | |
|---|---|---|---|---|
| $\mathcal{L}_1$ | $\mathcal{L}_s$ | $\mathcal{L}_{reg}$ | PSNR | SSIM |
| × | ✓ | × | 8.462 | 0.1533 |
| ✓ | × | × | 19.46 | 0.4952 |
| ✓ | ✓ | × | 26.68 | 0.7562 |
| ✓ | ✓ | ✓ | **26.94** | **0.7563** |

Table 3: Ablation study on input/output forms.

| Settings | | Metrics | |
|---|---|---|---|
| Input | Output | PSNR | SSIM |
| Spatial | Para. | 20.01 | 0.5169 |
| Spatial | Kernel↑ | 22.50 | 0.6585 |
| DFT | Para. | 20.53 | 0.5417 |
| DFT | Kernel↑ | **26.96** | **0.7576** |

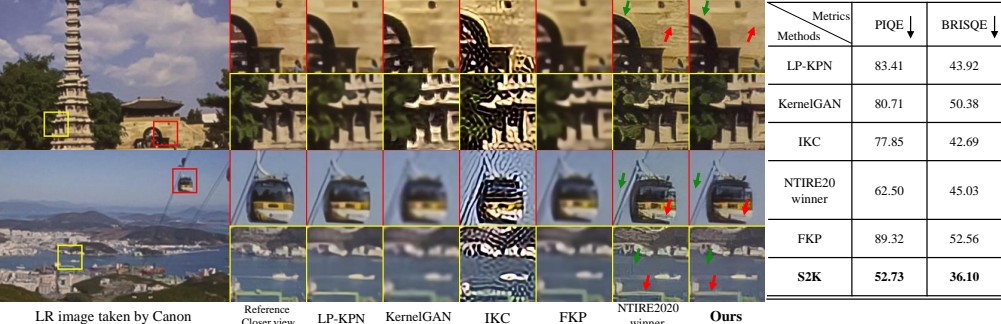

| Metrics / Methods | PIQE ↓ | BRISQE ↓ |
|---|---|---|
| LP-KPN | 83.41 | 43.92 |
| KernelGAN | 80.71 | 50.38 |
| IKC | 77.85 | 42.69 |
| NTIRE20 winner | 62.50 | 45.03 |
| FKP | 89.32 | 52.56 |
| **S2K** | **52.73** | **36.10** |

Figure 8: Visual comparison of $4\times$ blind SR results on real images taken by Canon from RealSR[4].

blind SR performance as shown in Figure 7. With the same model architecture, when the number of parameters decreases significantly, subsequent SR performance with kernel estimated in the frequency domain estimation is reduced by 1.55dB and 0.63db respectively, and for spatial domain is reduced by 9.45dB and 10.08dB respectively. The effective shape information provided in the frequency domain can still *ensure relatively good SR performance for the model with fewer parameters*. However, spatial estimation is more sensitive to the model size and capability.

**Ablation Study**    We conduct ablation study on loss components of the proposed generator. The results are shown in Table 2. We conduct extra comparisons with the same encoder network and training pipeline to verify the superiority of obtaining kernel directly from the frequency spectrum. Results can be found in Table 3. The ablation experiments are conducted on 2 additional test sets using the setting of $3\times$ blind SR on DIV2K valid set degraded with *random* Gaussian kernels (Para. and Kernel↑ briefly denote kernel parameters and up-sampling interpolation of kernel matrix).

Kernel estimation errors in different positions have unequal influences on subsequent SR results, $\ell_1$ without position information is inconsistent with SR results when the gap is small. $\ell_s$ allows the generator to learn the shape of the target kernel (*i.e.*, 2D-Gaussian curve). The smooth prior $\ell_{reg}$ is conducive to the initial generator's learning. Our method uses the translation from the spectrum to the interpolated kernel, which avoids the learning of local spatial pixel distribution, dramatically improving the learning stability. We achieve more superior kernel estimation accuracy and better SR results with the same network structure and pipeline compared with other settings.

## 4.2   Blind SR on Real Images

For real images, we use the paired image taken with a closer view in the dataset [4] as the content-fidelity reference and take into account the visual perception of the SR results. The comparison methods include KernelGAN [2], LP-KPN [4], IKC [6] and NTIRE20 winner [10]. We adopt the Gaussian kernel estimation setting with smaller variances and use the same SR pipeline as [10] but replace the kernel estimation part with our proposed S2K. The result of $4\times$ SR is shown in Figure 8. Compared with NTIRE20 winner [10], our results achieve higher fidelity (red indicators) and fewer artifacts (green indicators). There is no ground truth in the strict sense, and we provide the non-reference evaluation metrics for the real image SR results as Figure 8 right.

## 5   Conclusion

This paper theoretically demonstrates that feature representation in the frequency domain is more conducive for kernel estimation than in the spatial domain. We further propose S2K network, which makes the full use of the shape structure from the degraded LR image's frequency spectrum and, with

a feasible implicit cross-domain translation, outputs the corresponding up-sampled spatial kernel directly. In addition, we apply the kernel estimation optimization objective consistent with subsequent SR performance. Comprehensive experiments on synthetic and real-world datasets show that, when plugging S2K into the off-the-shelf non-blind SR models, we achieve better results both visually and quantitatively against state-of-the-art blind SR methods, by a large margin.

## Acknowledgments and Disclosure of Funding

This work is supported by the Natural Science Foundation of China under Grant 61672273 and Grant 61832008.

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
