# Supplementary materials

## 1 Proof of Eq. 8

$\phi(G, F) \leq \|\delta_\tau(F)\|_0 \ll \|\delta_{2\tau}(\mathbf{I}_D)\|_0 - \|\delta_\tau(\mathbf{k}_p)\|_0 \leq \phi(\mathbf{k}_p, \mathbf{I}_D)$

$$\phi(G, F) = \sum_{i,j} sign(\delta_\tau(G_{ij} - F_{ij})) \tag{1}$$

$$= \sum_{i,j} sign(\delta_\tau(F_{ij}S_{ij} - F_{ij})) \tag{2}$$

$$= \sum_{F_{ij} < \tau} sign(\delta_\tau(F_{ij}S_{ij} - F_{ij})) + \sum_{F_{ij} \geq \tau} sign(\delta_\tau(F_{ij}S_{ij} - F_{ij})) \tag{3}$$

$$\leq \sum_{F_{ij} < \tau} sign(\delta_\tau(F_{ij}(S_{ij} - 1))) + \sum_{F_{ij} \geq \tau} sign(\delta_\tau(F_{ij})) \tag{4}$$

$$= 0 + \sum_{F_{ij} \geq \tau} sign(\delta_\tau(F_{ij})) \tag{5}$$

$$= \|\sigma_\tau(F)\|_0 \tag{6}$$

$$\phi(k_p, I_D) = \sum_{i,j} sign(\delta_\tau([k_p]_{ij} - [I_D]_{ij})) \tag{7}$$

$$= \sum_{[k_p]_{ij} > \tau} sign(\delta_\tau([k_p]_{ij} - [I_D]_{ij})) + \sum_{[k_p]_{ij} < \tau} sign(\delta_\tau([k_p]_{ij} - [I_D]_{ij})) \tag{8}$$

$$\geq \sum_{[k_p]_{ij} < \tau} sign(\delta_\tau([k_p]_{ij} - [I_D]_{ij})) \tag{9}$$

$$\geq \sum_{[k_p]_{ij} < \tau, [I_D]_{ij} > 2\tau} sign(\delta_\tau([k_p]_{ij} - [I_D]_{ij})) \tag{10}$$

$$\geq \|\sigma_{2\tau}(I_D)\|_0 - \|\delta_\tau(k_p)\|_0 \tag{11}$$

where both $F$ and $k_s$ are sparse such that $\|\sigma_\tau(F)\|_0$ and $\|\delta_\tau(k_p)\|_0$ are significantly smaller than $\|\sigma_{2\tau}(I_D)\|_0$ due to the non-sparsity of $I_D$. So we come to $\phi(k_p, I_D) \gg \phi(G, F)$.

## 2 Fourier transform of 2D-Gaussian function.

(1) For the two-dimensional Gaussian function $f(x, y)$,

$$f(x, y) = \frac{e^{-(\frac{x^2}{\sigma_x{}^2} + \frac{y^2}{\sigma_y{}^2})}}{2\pi\sigma_x\sigma_y} \tag{12}$$

Submitted to 35th Conference on Neural Information Processing Systems (NeurIPS 2021). Do not distribute.

where $\sigma_x$ and $\sigma_x$ is the variance in two orthogonal directions respectively. After two-dimensional Fourier transform:

$$F(u,v) = \sum_x \sum_y f(x,y) e^{-j2\pi(ux+vy)} \tag{13}$$

$$= e^{-2\pi^2(\sigma_x^2 u^2 + \sigma_y^2 v^2)} \tag{14}$$

$$= A e^{-(\frac{u^2}{2\sigma_u^2} + \frac{v^2}{2\sigma_v^2})} \tag{15}$$

$$\sigma_u \propto \frac{1}{\sigma_x}, \sigma_v \propto \frac{1}{\sigma_y} \tag{16}$$

Where $\sigma_u$ and $\sigma_v$ denote the new variance of the Gaussian function after transformation. Discrete Fourier Transform (DFT) result $F_g(u,v)$ for *Gaussian kernel* is also in *Gaussian form*, and their variances are in *inverse proportion* [1].

# 3 Limitations

(1) As described in Section 3.1 of this paper, same with most previous blind SR methods, our method is also based on convolution and downsampling degradation model to describe the real degradation process, which is the commom setting. But this description may not cover all forms of degradation, such as non-uniform degradation (kernel each local area may be different) and motion synthesis using aliasing of adjacent frames. This needs further research and exploration in future work.

(2) Compared with the previous work, we provide a more accurate and efficient blind blur kernel estimation scheme. Combined with the existing efficient non-blind SR methods, we achieve the best blind SR results. Since the proposed estimation scheme is task independent and we mainly focus on blind-SR, it may also be suitable for some other blind task scenarios such as deblurring, which needs more future exploration.

# 4 More quantitative and visual results

In addition to the fidelity-oriented blind-SR experiment in the main text, we conducted additional perceptual-oriented experiments. Same as the previous experimental setup, we use ESRGAN and LPIPS perceptual metric to compare the results of $2\times, 3\times, 4\times$ blind-SR results on LR images from DIV2K degraded by random Gaussian kernels, as shown in the Table 1.

| Method | DIV2K | | | Flicker2K | | |
|---|---|---|---|---|---|---|
| | 2x | 3x | 4x | 2x | 3x | 4x |
| ESRGAN | 0.4969 | 0.5757 | 0.6315 | 0.4881 | 0.5719 | 0.6269 |
| FCA | 0.2799 | 0.3527 | 0.3818 | 0.2627 | 0.3488 | 0.3661 |
| KernelGAN | 0.2275 | 0.3159 | 0.5774 | 0.2371 | 0.3331 | 0.6141 |
| **Ours** | 0.1968 | 0.2569 | 0.3390 | 0.1987 | 0.2706 | 0.3400 |

Table 1: Quantitative [LPIPS↓] comparison of perception-oriented SR results for $2\times$, $3\times$, $4\times$ up-sampling respectively. The best performance is shown in red and the second best is blue.

Here we also provide a comparison of blind SR performance on the additional synthetic dataset L20 in Table 2. And more visual comparison with state-of-the-art methods as provided and shown in Figure 1, 2, 3 and Figure 5, we show the visual contrast of the *best* methods.

# 5 Code, data, and instructions needed to reproduction

We provide the code, instructions and dataset with download address for reproduction in the attached zip file.

---

[1]Reflect in the major and minor axis of the projection boundary on the position coordinate plane

| Method | Kernel | L20 | | |
| --- | --- | --- | --- | --- |
| | | 2x | 3x | 4x |
| RCAN finetuned | | 27.05 / 0.7310 | 24.78 / 0.6569 | 23.68 / 0.6220 |
| ZSSR | | 27.00 / 0.7308 | 24.76 / 0.6569 | 23.66 / 0.6222 |
| DeblurGAN w. RCAN | | 27.63 / 0.7516 | 25.30 / 0.6710 | 24.03 / 0.6301 |
| KernelGAN | | 27.39 / 0.7349 | 25.02 / 0.6649 | 24.38 / 0.6389 |
| FCA w.RCAN | G | 31.06 / 0.8607 | 28.07 / 0.7743 | 26.25 / 0.7107 |
| IKC | | 29.86 / 0.8670 | 28.80 / 0.7895 | 27.10 / 0.7348 |
| **S2K w. SFTMD** | | 32.26 / 0.8803 | 28.95 / 0.7909 | 27.51 / 0.7361 |
| **S2K w.RCAN** | | 32.73 / 0.8838 | 29.35 / 0.7939 | 27.83 / 0.7355 |
| RCAN finetuned | | 24.88 / 0.6775 | 23.04 / 0.6158 | 22.02 / 0.5906 |
| DeblurGAN w. RCAN | | 24.77 / 0.6830 | 23.03 / 0.6179 | 21.90 / 0.5897 |
| ZSSR | | 24.85 / 0.6756 | 23.02 / 0.6151 | 22.02 / 0.5903 |
| KernelGAN | M | 24.16 / 0.6365 | 22.85 / 0.6057 | 21.10 / 0.5418 |
| IKC | | 27.57 / 0.7901 | 23.53 / 0.6575 | 22.60 / 0.6187 |
| Pan *et al.* w.SFTMD | | 22.63 / 0.6312 | 21.19 / 0.5647 | 19.84 / 0.5411 |
| **S2K w. SFTMD** | | 32.95 / 0.9070 | 30.92 / 0.8551 | 28.36 / 0.7842 |

Table 2: **Quantitative [PSNR↑ / SSIM↑] comparison results of fidelity-oriented SR model for $2\times, 3\times, 4\times$ up-sampling.** G, M denote Gaussian kernels and motion kernels respectively. The best performance is shown in red and the second best in blue.

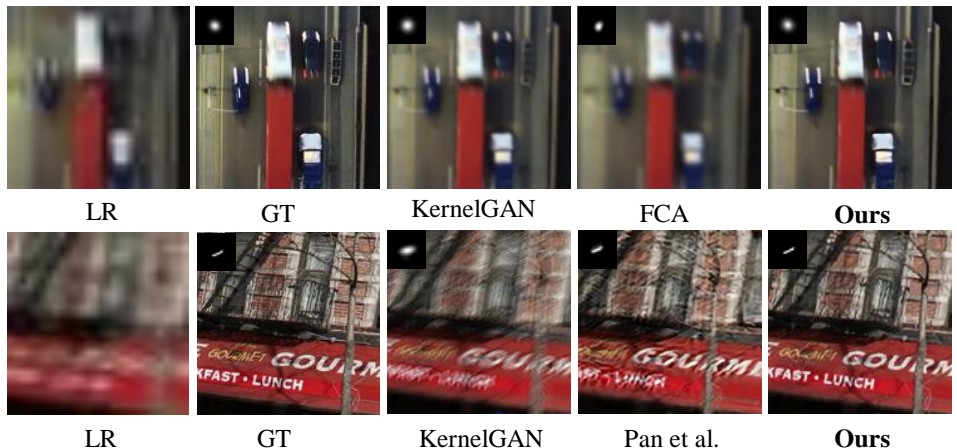

Figure 1: $2\times$ blind SR results for LR degraded with unknown kernel from DIV2K. For FCA is not able to handle motion kernel, so we use Pan *et al.*'s as a comparison instead. Due to more accurate kernel estimation, our method achieves the most pleasant results compared with other methods.

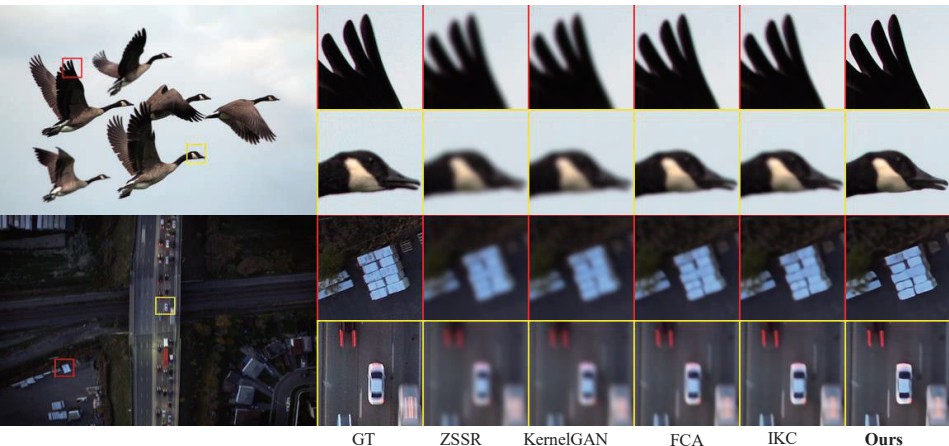

Figure 2: $2\times$ blind SR results for LR degraded with unknown Gaussian kernel from DIV2K.

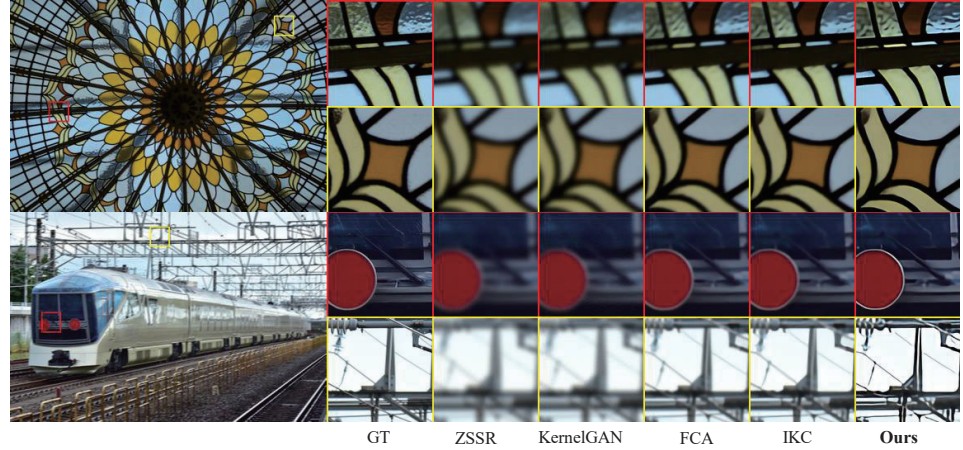

Figure 3: 2× Blind SR results for LR degraded with unknown Gaussian kernel from DIV2K.

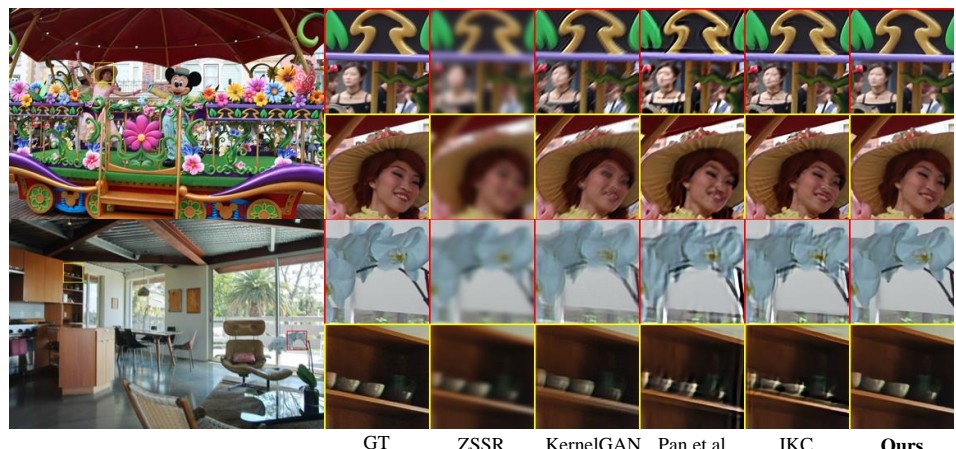

Figure 4: 2× Blind SR results for LR degraded with unknown motion kernel from Flicker2K.

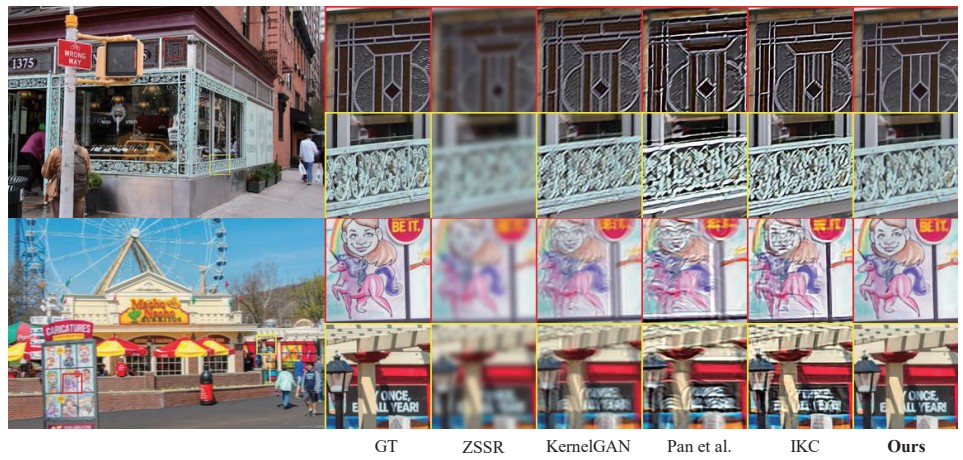

Figure 5: 2× Blind SR results for LR degraded with unknown motion kernel from Flicker2K.