# OpenReview forum: "Spectrum-to-Kernel Translation for Accurate Blind Image Super-Resolution"
_NeurIPS.cc/2021/Conference — NeurIPS 2021 Poster_

### Official Review · Reviewer_bKSR · 2021-07-15

**Rating:** 7
**Confidence:** 4

**Summary:**

The authors make a key observation that degradation kernels are highly correlated to degraded images in the frequency domain. From this, they design a kernel generator that directly generates the degradation kernel from the frequency-domain LR image. Then, a non-blind SR model is employed to utilize the estimated kernels for SR. The proposed method outperforms existing methods on two types of kernels (Gaussian and motion) with two backbone networks (SFTMD and RCAN).

**Limitations And Societal Impact:**

Limitations are discussed in the supplementary material but potential negative societal impact has not been addressed. Some potential negative societal impacts could include privacy issues due to upscaling degraded data.

**Main Review:**

Pros:
- This paper provides insightful observations that degradation kernels are highly correlated to the degraded image in the frequency domain (Fig. 1 and Fig. 2).
- From the above observation, a novel approach is taken, where the blur kernel is directly estimated from the frequency-domain LR image.
- A sound theoretical analysis is provided in Section 3.1 .
- Supplemetary material provides additional details that helps to further understand the main paper.
- Ablation studies verify the benefits of each of the proposed components including the spectrum-to-kernel scenario.
- The proposed method outperforms existing methods on real images.

Cons:
- Clarifications are needed in some parts of the paper:
  - The phrase in lines 37-39, “the iterative form can not utilize the maximum performance of both estimate and SR module during the testing process, and the barrel effect is inevitable", is unclear. Why can't IKC reach maximum performance for both the estimator and the restorer? What do the authors mean by barrel effect?
  - What do the authors mean by "inconsistency of training and test domain" in lines 216-217? Aren’t kernels generated randomly for both training and testing?
  - In line 208, what do the authors mean by 'random blurred'? Is this random Gaussian blur or randomly selected motion blur kernels?
- Some missing analyses in experiments:
  - Is there a reason why the performance gain reduces as scale factor increases?
  - Why do existing methods perform poorly for motion kernels? Aren’t they trained on the same random motion kernels used for training the proposed model?
- In Table 2, comparison to more recent blind SR methods such as DAN (NeurIPS 2020) is missing.
- It is common to include SR results on historic images (http://vllab.ucmerced.edu/wlai24/LapSRN/) as in ZSSR, IKC, etc in blind SR literature.
- The motivation behind bicubic downsampling the FFT image before the kernel generator is unclear. Wouldn't this lose information? If it is to reduce computations, why not downsample by 4 or 8?
- Statements like “existing blind SR methods are far from meeting the needs of various degradation processes in general scenes” (lines 30-31), need references or evidence. How are they ‘far from meeting the needs of various degradation’?

Minor:
- In Table 4, what is the meaning of para.?
- It would be useful to show LR input images in Fig. 5.
- Typos: (i) Table 1 caption, estimaion -> estimation, (ii) line 234, we -> We.

========================
**Post Rebuttal:**
Raising my score from 6 to 7 after the authors' reply. More details are in a separate comment. Well done on the new experiments and the clarifications!

**Time Spent Reviewing:**

3

---

> ### Author Response · Authors · 2021-08-10
> **Response to Reviewer bKSR.**
>
> Q1. Why can't IKC reach maximum performance for both the estimator and the restorer?  barrel effect?
>
> A1. The expression of bucket effect is not rigorous enough. Actually we  want to express that IKC is a method based on Maximum a posteriori estimation ($MAP$), in which the estimator depends on the posteriori of the pre-trained restorer. Due to the accuracy limitation of the restorer, the maximum performance of the estimator cannot be guaranteed. As demonstrated in [R-1], Levin et al. explained the failure of the naive $MAP$ approach by demonstrating that it mostly favors no-blur explanations, and  proofed the deconvolution result $x$ and kernel $k$ in optimizing the $MAP_{x,k} $  score satisfies $|x| \rightarrow 0 $ and $|k| \rightarrow \infty$ (Claim 1). Accordingly, we claim estimating kernel independently is more practical, effective and expansible. We will fix this vague expression and make it more clear in revision.
>
> Q2. "Inconsistency of training and test domain" in lines 216-217? Aren’t kernels generated randomly for both training and testing?
>
> A2. Basically, for all methods, training HR images are clear, while test LR images are blurred by unknown random blur kernels from a wide range. More specifically, the blind-SR methods integrate kernel information into the training process. As the result, the more precise the kernels are estimated from test LR images, the better blind SR results we can achieve. However, non-blind methods (e.g., RCAN, ZSSR) are not able to estimate kernel, thus can only obtain LR images based on bicubic downsampling in the training process, which leads inconsistency between training (bicubic) and test (random kernels) domains.
>
> Q3. In line 208, what do the authors mean by 'random blurred'?
>
> A3. Sorry for the unrigorous description. Actually, we use different blur kernels for different settings. As shown in Tab. 2, we perform the random Gaussian kernels for the Gaussian setting, and the random motion kernels for the motion setting.
>
> We further explore the effect of mixed blur kernels (i.e., Gaussian, motion and disk kernels) during training on our S2K w. SFTMD. The blind SR results on the test images blurred wth only Gaussian kernel or motion kernel (same setting in Tab. 2) are as follows. Compared with the model trained by certain type of kernel, the model trained by mixture kernels is slightly worse, but is able to deal with more types of kernels by a single model. We will include the above analysis in revision to make it more rigorous.
>
> | | **DIV2K** | **Flicker2K** |
> |:-:|:-:|:-:|
> | Gaussian kernels| 25.65 | 25.64|
> | Motion kernels| 26.76 | 26.53|
>
> Q4. Is there a reason why the performance gain reduces as scale factor increases?
>
> A4. The performance of blind-SR is mainly affected by the SR process and kernel estimation process, in which the kernel estimation from frequency domain is less affected by the scale due to the characteristic of our degradation model. However, for SR model, it becomes more difficult with the increase of scale. This phenomenon is also commonly oberserved in  previous SR methods.
>
> Q5. Why do existing methods perform poorly for motion kernels? Aren’t they trained on the same random motion kernels used for training the proposed model?
>
> A5. We ensure that all kernel-aware methods use the same training and test settings for fairness. The reason for the poor performance of the existing methods on motion kernel is that most existing blind SR methods only aim at predicting Gaussian kernels without considering motion kernels. Gaussian kernels, equivalent to low-pass filtering, with fixed form, few parameters and easy dimensionality reduction cause regular blur. On this basis, most methods propose specific pipeline with Gaussian kernel prior such as blur degree mismatch [R-3], frequency domain consistency [R-4] and reversible  projection [R-5], while these schemes are not suitable for motion kernel. Although some researches consider the motion kernel in blind setting, due to weak estiamtion prior and spatial pipline, the performance is limited even with handcraft constraint [R-2] including kernel's sparsity constraint, boundary constraint (referring to Gaussian kernel) and down sampling constraint (referring to bicubic kernel), etc.
>
> Q6. In Tab. 2, comparison to more recent blind SR methods such as DAN (NeurIPS 2020) is missing.
>
> A6. Extra 4x SR experiments on 2 detaset degraded with random anisotropic Gaussian kernels are supplemented as follows, in which we retrained DAN with exactly the same setting of Gaussian kernel conducted in Tab. 2.
>
> | | **DIV2K** | **Flicker2K** |
> |:-:|:-:|:-:|
> | DAN(NeurIPS20)| 25.68 | 25.70|
> | S2K w. SFTMD| ***25.84*** | ***25.79***|
>
> DAN also adopts an alternative optimization algorithm and corrects the estimator according to the results of the restorer in the spatial domain. Benefiting from the significant advantages of frequency domain estimation kernel, we can directly obtain an accurate kernel and make full use of the performance of estimation and SR process, which accounts for our better performance.
>
> Q7. Non-reference evaluation comparison of 4x blindSR on real history images.
>
> | | **ZSSR** | **KernelGAN** |**IKC**|**S2K w.RCAN**|**S2K w.ESRGAN**|
> |:-:|:-:|:-:|:-:|:-:|:-:|
> | PIQE ↓|83.05|70.77|41.98|**36.67**|***26.1195***|
> | BRISQE↓| 47.10|43.96|	34.29|**30.70**|***20.3382***|
>
> A7. Per your request, we conduct a test on 10 real-world history images to demonstrate the effectiveness of our S2K compared with several SOTA blind SR methods. An extra method Blind-SR-SA (Blind Image Super-Resolution with Spatially Variant Degradations, Siggraph Asia 2019) mentioned by Reviewer idYk is also included for comparison. Specifically, since official IKC only provides a 4x SR pre-trained model (on DIV2K+Flicker2K), and Blind-SR-SA only provides a 2x SR pre-trainied model (on DIV2K), we compare our method (on DIV2K, without noise injection) with IKC and Blind-SR-SA on 4x and 2x SR settings, respectively. Without violating the principle of anonymity, we provide some links to visual comparison results.
>
> img008: https://imgsli.com/NjQ3MjU  (4x SR result  S2K w. ESRGAN vs. Official IKC)
>
> img010: https://imgsli.com/NjQ3MjA  (4x SR result  S2K w. ESRGAN vs. Official IKC)
>
> img007 : https://imgsli.com/NjQ3MjE  (4x SR result  S2K w. ESRGAN vs. Official IKC)
>
> img001: https://imgsli.com/NjQ4MjU   (2x SR result  S2K w. RCAN vs. Official Blind-SR-SA)
>
> img004:  https://imgsli.com/NjQ4MjY  (2x SR result  S2K w. RCAN vs. Official Blind-SR-SA)
>
> img009: https://imgsli.com/NjQ4Mjg  (2x SR result  S2K w. RCAN vs. Official Blind-SR-SA)
>
> As we can see, the results of IKC and Blind-SR-SA are still relatively blur, while our S2K can recover better structure and edge in the image with less artifact and noise. For example, our S2K effectively restores the flags, trees and other details in img008, the architectural structure outline in img010, the trains and sleepers in img007, and the characters and people's clothes in img001, etc.
>
> Q8. The motivation behind bicubic downsampling the FFT image before the kernel generator is unclear. Wouldn't this lose information? If it is to reduce computations, why not downsample by 4 or 8?
>
> A8. The purpose of downsampling 2x is a trade-off to reduce the amount of calculation without causing less information loss and keep consistent with the input size of original PIX2PIX network.
>
> Q9. Statements like “existing blind SR methods are far from meeting the needs of various degradation processes in general scenes” (lines 30-31), need references or evidence. How are they ‘far from meeting the needs of various degradation’?
>
> A9. Sorry for the unrigorous statement. In fact, we want to emphasize that previous blind SR methods is not degradation-aware enough on real scene images, resulting in a gap between the SR results and human visiual expectation [R-2], and the involving kernel is relatively simple [R-6]. We can achieve significant performance improvement under more general degradation settings. Besides, the complexity of several methods also leads to their limited application in practical [R-4]. Our S2K can flexibly provide accurate kernel estimation for the existing non blind methods.
>
> Q10. Minor issues:
>
> A10.
>
> - 'Para.' in Tab. 4 indicates estimating kernel in the form of a number of parameters.
>
> - Sure, we will include the LR images in Fig. 5 in revision.
>
> - Thanks, we will revise the typos in revision.
>
> [R-1] Anat Levin et.al., Understanding and evaluating blind deconvolution algorithms., In: PAMI, 2011.
>
> [R-2] Sefi Bell-Kligler et.al. Blind super-resolution kernel estimation using an internal-gan.,  In NeurIPS 2019.
>
> [R-3] Jinjin Gu et.al., Blind super-resolution with iterative kernel correction., In: CVPR 2019.
>
> [R-4] Xiaozhong Ji et.al.,  Frequency consistent adaptation for real world super resolution., In AAAI 2021.
>
> [R-5] Jingyun Liang et.al., Flow-based kernel prior with application to blind super-resolution., In: CVPR 2021.
>
> [R-6] Ruofan Zhou et.al. Kernel modeling super-resolution on real low-resolution images., In ICCV, 2019.

---

> ### Comment · Reviewer_bKSR · 2021-08-23
> **Most of my concerns were addressed**
>
> My biggest concern with this paper was its clarity. The authors clarified some parts in their reply and promised to make revisions, so I expect this to improve the readability of the paper.
>
> Another concern was related to experiments. I appreciate the additional experiments provided by the authors regarding (i) mixed blur kernels, (ii) comparison to a more recent blind SR method, and (iii) quantitative and qualitative results on real historic images, which would all be nice additions to the paper. The overall visual results on historic images are impressive, although in some cases (img010), there are artifacts that seem to be produced due to GAN which is understandable. The generalization ability and the overall performance of the proposed method became much more convincing with these additional results, and the authors also mentioned that the compared methods were trained under the same setting.
>
> I really like the simple idea of predicting the degradation kernel in the frequency domain, and I'm convinced that the proposed method works well on a variety of settings. Although some concerns on overall clarity remain, I will happily raise my score to 7, expecting the authors to make revisions regarding Q1, 3, 5, 6, 7, 9 and 10.

---

### Official Review · Reviewer_idYk · 2021-07-16

**Rating:** 6
**Confidence:** 4

**Summary:**

The authors propose a novel framework for solving the problem of blind super resolution in which the low resolution input can be degraded by arbitrary blur kernels. In their framework, the degradation kernel estimation is performed in the frequency domain. The authors show that the frequency domain is better suited for kernel representation and reconstruction than the spatial domain and propose a Spectrum-to-Kernel mapping network to estimate diverse blur kernels. They use a conditional GAN to learn how to 'translate' the spectra of degraded images to the unknown kernels.

**Limitations And Societal Impact:**

I would not expect negative societal impact from this work. The limitations are listed in the supplemental material but might actually deserve a place in the paper. At least the point regarding spatially varying degradations.

**Main Review:**

Estimating degradation kernels for the case of blind upscaling is typically an ill posed and difficult task and estimation errors can carry over into conditional upscaling methods resulting in artefacts in the upscaled image. Therefore, efforts to better represent and estimate degradations automatically is relevant.

The idea of estimating kernels in a different representation seems nice and the authors demonstrate that for commonly existing sparse kernels, reconstruction in the frequency domain is preferable compared to the spatial domain. The observation that there is a closer resemblance between the frequency spectrum of the kernel and the frequency spectrum of the image degraded with that kernel as compared to the spatial kernel and the degraded image in the spatial domain is interesting. Also as the authors state, I am not aware that this has been leveraged for the task of blind super resolution so far. Therefore it seems intriguing to design a network to estimate a kernel from the Fourier amplitude spectrum. Also the results shown look competitive.

However, in my opinion the work [CORNILLÈRE et al., Blind Image Super-Resolution with Spatially Variant Degradations, Siggraph Asia 2019] is quite related and should be cited and compared to. There, the estimation of the degradation is based on an optimisation that discriminates artefacts in the resulting high res while being parameterised through a neural network embedding. Fundamentally, it seems like a very different approach to estimate the degradation in the low res. A lot of existing works follow the path of estimating the degradation from the low res and here I can see that it could be better solved int the frequency domain. However, in the resulting high res where kernel estimation errors quickly become very visible in the form of ringing/oversharpening or excessive blurring, it seems that also methods operating in the spatial domain might find accurate solutions. Therefore I would like to see comparisons as mentioned before and ideally a small paragraph to interpret the results and differences in the approaches.

Overall Equation 8 seems to indicate that the frequency domain might be better for estimating degradation kernels but I'm not sure it can directly be derived from the equation since it should only show that kernel and degraded image are closer together w.r.t. a given metric in frequency space as opposed to image space. While this gives a good motivation to follow that strategy, it seems that further assumptions/ restrictions would be needed to conclude that this will also allow more accurate kernel estimation / reconstruction. It would be great if the authors could clarify that a bit more. As a more minor point, it would also be great to detail a bit more about how reconstructing the kernel in the spatial domain would compare to reconstructing in the frequency domain.

Overall, I feel this paper offers an interesting view on blind image super resolution and tries a novel approach that has not been analysed so far in this context. However at the same time some important comparisons are missing and the paper seems slightly unpolished with a larger number of typos.

**Time Spent Reviewing:**

2-3 hours

---

> ### Author Response · Authors · 2021-08-10
> **Response to Reviewer idYk.**
>
> Q1. Whether it has been used for blind SR tasks.
>
> A1. To our best knowledge, we are the ***first*** to learn the translation from degraded LR input to blur kernels entirely in frequency domain. There have been several frequency-based methods previously, but they either roughly divide spatial LR input into low/high-frequency components with pixel/adaptive losses respectively, or use overall frequency  distribuction as a consistency constraint for spatial learning. We abandon the spatial pixel distribution characteristics, and instead use the properties of Fourier transform and all the information of spectrum. We theoretically deduce the better learning habit from FFT spectrum to kernel. Then, extensive experiments show that the proposed method can provide more accurate estimation results than spatial methods or other frequency domain analysis based methods.
>
> Q2. Comparison with [CORNILLÈRE et al., Blind Image Super-Resolution with Spatially Variant Degradations, Siggraph Asia 2019].
>
> A2. We supplemented the quantization and visual comparison of historical images with CORNILLÈRE' work (denoted as Blind-SR-SA) in the reponse to ***Reviewer bKSR.A7*** along with several other SOTA blind SR methods. In conclusion, our S2K achieves much better results both quantitatively and qualitatively.
>
> Specifically, Similar to IKC(CVPR19) and DAN(NeurIPS20), Blind-SR-SA is also a MAP-based method. In general, the main difference is that we estimate completely in the frequency domain and take the estimation kernel as an independent task. In the experimental setup, we include not only Gaussian kernels and disk kernels in a wide range, but also more irregular motion kernels. We will add the comparison with this method and the comparison of quantitative visual results on real historical images in revision.
>
> Q3. More details about how reconstructing the kernel in the frequency domain.
>
> A3. Per your request, we provide the visualization results of the intermediate features in the form of heatmaps https://imgsli.com/NjUwMjM, illustrating the process of the input frequency spectrum being translated to the kernel (please refer to ***Reviewer ZgQH.A1*** for more details). Currently, our method mainly focuses on improving the prediction of uniform kernels. It may be a good future direction to expand our framework to deal with more practical non-uniform kernels.

---

### Official Review · Reviewer_ZgQH · 2021-07-24

**Rating:** 6
**Confidence:** 5

**Summary:**

This paper proposes a new blind super-resolution method, which can due with arbitrary blur kernels instead of Gaussian kernels. The whole method estimates kernels in the frequency domain. The method involves a kernel generator that takes the frequency amplitude as input and outputs a kernel. A patch discriminator is designed to perform adversarial loss to the predicted kernel. This design is novel and the experiments show good results.

**Ethics Review Area:**

["I don’t know"]

**Limitations And Societal Impact:**

Yes

**Main Review:**

I have several questions:

1. Can we know how this kernel was predicted? Is there any visualization method that can tell us which part of the frequency amplitude contributes to the kernel prediction? This is important when it comes to the generalization performance of the proposed method. Although the experiment shows good generalization performance in Figure 6, an interpretational explanation is also wanted to address our concerns about how this system works.

2. As a common sense that the kernel prediction is an ill-posed problem. How does the proposed method address this issue?

3. This paper actually proposes a kernel estimation method. Can this method be used for deburring motion blur? How is the performance? If it is specially designed for blind super-resolution, what makes it especially suitable for SR task?

4. In Table 2, the motion blur part, The IKC, Pan et al. [22] w. SFTMD, Pan et al. [23] w. SFTMD and other methods are not trained for motion blur, the author should compare their method with motion-deblur + regular SR for better results. The claim of nearly 6db improvement is not objective as the STOA performance on these images is not obtained by the abovementioned methods.

As the good performance and novel idea (predict in frequency domain), I give a positive initial rating. I hope the authors can answer my questions so that I can make the final suggestion.

**Time Spent Reviewing:**

5

---

> ### Author Response · Authors · 2021-08-10
> **Response to Reviewer ZgQH.**
>
> Q1. Interpretability of kernel estiamtion.
>
> A1. We take the channel with the largest variance from the feature map output by the middle layer of the model, and draw a heat map for visualization. These results show that the shape of the input spectrum is critical in S2K estimation (the edges are extracted by the middle layer of the network), which verifies the claim of L156-157 in the paper.
> Visualization results are provided at https://imgsli.com/NjUwMjM. In decoder, the rectangle marked by the red box shows the learned structural shape provided by the spectrum.
>
> Q2.  Ill-poseness of kernel prediction.
>
> A2. The training of S2K network provides kernel's GT. We construct a large sample training set covering three kernel types to learn the translation from the blurred image spectrum to the kernel in the frequency domain. Our assumption for kernel estimation is that the original image is high-definition (like DIV2K), so all elements in the degradation process correspond one by one, which alleviates the ill-posed problem of kernel estimation to a certain extent.
>
> Q3. This paper actually proposes a kernel estimation method. Can this method be used for deblurring motion blur? How is the performance? If it is specially designed for blind super-resolution, what makes it especially suitable for SR task?
>
> A3.  In general, our method can not be directly used for deblurring motion blur task because of the different targets of the two kinds of tasks. Specifically, current deblurring methods mainly focus on non-uniform motion blur, in which the blurred image is generated by averaging consecutive and clear sequential frames. In contrary, blind SR methods, including our S2K, mainly focus on addressing uniform kernels, and the popular benchmarks in blind SR do not satisfy the requirement of sequential data.
>
> Although our method can not directly compare with the recent deblurring methods, such as MPRNet(CVPR21), we perform the following setting for uniform motion blur to compare MPRNet with ours. To be specific, we train a kernel-aware SR model SFTMD (7.97M parameters) with scale factor 1 to conduct the deblurring task.  We then train methods with 2$\times$ down-sampling of DIV2K training/validation as the target and use the mixed kernel blurred result as the input, and compare with the latest deblurring method MPRNet (20.13M parameters). Under the deblur setting of degraded modeling including blur kernel convolution, we can achieve higher results than the SOTA deblurring method by making full use of kernel prior and fusion in network training. We report PSNR comparison here:
>
> | **Method** | **S2K w. SFTMD** | **MPRNet(CVPR21)** |
> |:-:|:-:|:-:|
> | PSNR (db) |***32.43*** | 28.91|
>
> Q4. Motion-deblur + regular SR
>
> A4.  We compared the results with DeblurGAN w. RCAN in Table 2, the DeblurGAN are trained under the setting of bicubic down-sample LR image $x$ as target and blurring $x$ with gaussian/motion kernel as input $y$, but the effect is not good.
> In addition, under the same settings, we provide the 4$\times$ SR comparison on random motion blurred test set with the latest SOTA deblurring method MPRNet + regular SR SRResnet (SRResnet is SFTMD's backbone without kernel fusion).
>
> | **Method** | **S2K w. SFTMD** | **MPRNet w. SRResnet** |
> |:-:|:-:|:-:|
> | PSNR (db) | ***26.95*** | 23.01|
>
> Further, we argue the scheme of deblurring first and then regular SR is not ideal in blind SR setting for the following reasons:
> (1) The degradation process in the deblurring is different from the SR, as described in A3.
> (2) The scale of low resolution images is small. which is difficult to obtain  sufficient context for deblurring.
> (3) The image obtained by using the existing deblurring methods may not satisfy distribution obtained by bicubic downsampling.
> (4) Degradation separation may cause the deblurring error amplifying in the subsequent over-separation process.
> The main reason why our method achieves significant improvement over SOTA method on irregular motion kernel is making the best of degradation model prior and obtaining accurate blur kernel almost consistent with GT in frequency domain.
>
> [R-1]: Liang Chen et.al., Blind Image Deblurring With Local Maximum Gradient Prior., In: CVPR, 2019.
>
> [R-2]: Liyuan Pan et.al., Phase-Only Image Based Kernel Estimation for Single Image Blind Deblurring., In: CVPR, 2019.
>
> [R-3] Orest Kupyn et.al., DeblurGAN: Blind Motion Deblurring Using Conditional Adversarial Networks., In: CVPR, 2018.
>
> [R-4] Syed Waqas Zamir et.al., Multi-Stage Progressive Image Restoration., In: CVPR, 2021.

---

### Official Review · Reviewer_wMvH · 2021-08-02

**Rating:** 6
**Confidence:** 4

**Summary:**

This work proposes a method for blind super-resolution (SR) of images degraded by arbitrary blur kernels. The method makes use of a novel blur kernel estimation technique, that proposes to estimate such kernels in the frequency domain following an end-to-end deep learning approach. The authors claim that the Fourier amplitude spectrum kernel representation is better posed for the learning task of kernel estimation than the spatial representation, under the assumption that the kernels are sparse both in frequency and space domains (this assumption holds form most common kernels in real settings). They support this claim with theoretical and experimental evidence.

The image formation model considered here, that produces the degraded LR image from the HR image, is the following. The clean high-resolution (HR) image first downsampled, then convolved with the blur kernel, and finally contaminated with additive noise.

Based on the proposed frequency domain representation, the authors propose a kernel generator network that allows to translate this representation to the kernel spatial representation. The generator $G$ takes as input the single-channel amplitude spectrum (the FFT) of the degraded LR image bi-cubicly downsampled by two, and outputs the single-channel blur kernel estimate. This generator consists of an encoder-decoder network built on a U-net with mirror connection between each of the seven down-sampling convolutional layers and up-sampling transposed convolutional layers with the same feature size.

To ensure that the kernel generated by $G$ is at the same time close to the ground truth kernel and consistent with the SR results, they propose a three-terms loss for $G$. The first term imposes similarity between the ground truth kernel and the kernel generated by G from the degraded image (in the spatial domain) by means of the $\ell_1$; the second term also imposes similarity between these two kernels but in seen as probability density functions, using an adversarial loss that involves a trainable critic $D$ that has to  discriminate between the ground truth kernel and the output of G (actually, de concatenation of each of them with the input of G are considered instead). This patch discriminator is trained using the loss proposed in LSGAN. Finally, the third term is a regularization loss on the output of G; unless I missed it, the regularizer is not specified. This kernel estimation model is called S2K.

Then, the authors propose a pipeline for blind SR that combines the S2K module with a non-blind SR module. For the non-blind SR module, they consider two methods:  the multiple degradation-based method SFTMD (Gu et al., CVPR 2019; ref 6) and the kernel modeling-based method RCAN (Zhang et al., ECCV 2018; ref. 35). Given a ground truth kernel of size 15x15 and a HR image, a LR degraded image is generated and cropped to 512x512. The input to the S2K module is the ground truth kernel upsampled to 256x256 and the FFT of the LR image, downsampled to 256x256. The estimated kernel of size 256x256 returned by G is then downsampled at size 15x15. Finally, the non-blind SR module is fed with the LR degraded image and the 15x15 kernel estimated with the S2K module.

The experiments are carried out considering Gaussian kernels of size 15x15 of random covariance matrix, and random motion kernels of size 23x23 (instead of 15x15) with Boracchi and Foi’s method (IEEE TIP, 2012; ref 3). The proposed approach is compared to an extensive list of state-of-the-art methods, both for the task of blur kernel estimation and blind SR for 2x, 3x and 4x SR. All methods were trained on the the same degraded dataset (DIV2K), including S2K and the non-blind methods used in the proposed blind SR pipeline. Blind SR methods are compared on synthetic images using the PSNR and SSIM metrics. Kernel estimation performance is evaluated using the $\ell_1$ norm and KL divergence. The results reported by the authors show that the proposed S2K and the blind SR pipeline significantly outperform the others. Finally, an ablation study on the loss components and frequency vs. spatial inputs is conducted, showing the pertinence of the proposed approach.

Finally, blind SR experiments on real images using the dataset proposed by Cai et al. (ICCV 2019; ref. 4), evaluated using two reference-free metrics (PIQE and BRISQUE) show that the proposed pipeline also outperforms its competitors, including the NITRE20 winner (Ji et al., CVPR 2020; ref. 10).


**Limitations And Societal Impact:**

I do not see any limitation.

**Main Review:**

(+) The method is sound, elegant and simple

(+) References and review of previous work are adequate.

(+) Extensive evaluation, both on blur kernel estimation and blind SR.

(+) According to the results reported by the authors, the proposed approaches for blur kernel estimation and the blind SR tasks both outperform state-of-the-art methods by significant margins.

(+) Code is provided as supplementary material (I did not check it).

(-) The image formation model is not realistic. The way LR images are produced from HR images is by first convolving with the HR blur kernel, and then subsampling. Not the other way round.

(-) The fact that comparison of the spectrums of the degraded image with the blur kernel is more conducive than comparison of the degraded image and the blur kernel in space is evident. As the blur kernel low-pass filter the image, it is obvious that the coarse features of the spectrum of the kernel and of the degraded image are similar.

(-) Frequency consistency was already proposed and demonstrated in ref. 11 (not published but reported in arXiv). Moreover, the kernel estimation method proposed in ref. 11 bears strong similarities to the one proposed here. This works even proposes an adversarial loss for the high-frequency components of the blur kernels. The authors should clearly state the differences and contributions of their work compared with ref. 11.

(-) Last but not least, the paper is poorly written. While the idea is simple, the paper does not read well. For instance:

* When the authors show that comparison of the degraded image and the kernel in the frequency domain is more conducive than in the spatial domain, one wonders why this comparison is relevant. They only show its relevance with an example using a Gaussian kernel afterwards.

* Notation is sometimes misleading; for instance, in Section 3.2 $\mathbf{S}$ is used to represent the Fourier transform of the clean image, and then from Section 3.3 it is used to represent the spectrum of the degraded image.

* The methods presented in the tables should also be referenced in the tables.

* English is poor, both grammatically and syntactically. The text is plagued with conjugation errors, unstructured sentences and typos. The article should be almost completely rewritten and checked by someone fluent in English.


**Time Spent Reviewing:**

6

---

> ### Author Response · Authors · 2021-08-10
> **Response to Reviewer wMvH.**
>
> Q1. Degradation model.
>
> A1. Actually, we adopted the ***same*** degradation model, which is also recognized as a practical solution, used in several previous SOTA SR methods, including Eq. (2) in Learning a Single Convolutional Super-Resolution Network for Multiple Degradations (CVPR18), Eq. (3) in Deep Plug-and-Play Super-Resolution for Arbitrary Blur Kernels (CVPR19), and Eq. (1) in Frequency Consistent Adaptation for Real World Super Resolution (AAAI21), etc.
>
> Besides, the results on real-world images also demonstrated our S2K model is realistic. As shown in Fig. 8, our results of blind-SR on real images achieve the best visual effects and non-reference evaluation metrics performance as well. Furthermore, we find our method is still effective under the degradation using convolution with HR blur kernel first and then downsampling with scale factor as stride, which benefits from the strong advantages of frequency domain features. Estimated Gaussian kernels (zoom in to 128$\times$128 for better view) are provided at https://imgsli.com/NjUwMjA.
>
> Last but not least, we additionally illustrate the visual results of our method against two SOTA methods IKC(CVPR19) and Blind-SR-SA(Siggraph19) on several real-world history images. More details on experimental settings and quantitative results can also be found in ***Reviewer bKSR.A7***. Without violating the principle of anonymity, we provide some links to visual comparison results.
>
> img008: https://imgsli.com/NjQ3MjU  (4x SR result  S2K w. ESRGAN vs. Official IKC)
>
> img010: https://imgsli.com/NjQ3MjA  (4x SR result  S2K w. ESRGAN vs. Official IKC)
>
> img007 : https://imgsli.com/NjQ3MjE  (4x SR result  S2K w. ESRGAN vs. Official IKC)
>
> img001: https://imgsli.com/NjQ4MjU   (2x SR result  S2K w. RCAN vs. Official Blind-SR-SA)
>
> img004:  https://imgsli.com/NjQ4MjY  (2x SR result  S2K w. RCAN vs. Official Blind-SR-SA)
>
> img009: https://imgsli.com/NjQ4Mjg  (2x SR result  S2K w. RCAN vs. Official Blind-SR-SA)
>
> As we can see, the results of IKC and Blind-SR-SA are still relatively blur, while our S2K can recover better structure and edge in the image with less artifact and noise. For example, our S2K effectively restores the flags, trees and other details in img008, the architectural structure outline in img010, the trains and sleepers in img007, and the characters and people's clothes in img001, etc.
>
> Q2. Evident and simple
>
> A2. Thanks for appreciating our simple, novel yet effective idea, which is also well recognized by all of the other reviewers.
>
> To our best knowledge, we are the ***first*** to learn the translation from degraded LR input to blur kernels entirely in frequency domain. There have been several frequency-based methods previously, but they either roughly divide spatial LR input into low/high-frequency components with pixel/adversarial losses respectively, or use overall frequency distribution as a consistency constraint for spatial learning. We abandon the spatial pixel distribution characteristics, and instead use the properties of Fourier transform and all the information of spectrum. We theoretically deduce the better learning habit from FFT spectrum to kernel. Then, extensive experiments show that the proposed method can provide more accurate estimation results than spatial methods or other frequency domain analysis based methods.
>
> Q3. Differences with FCA.
>
> A3. Actually, we have made some comparisons with FCA in L39-L42, L92-L94 and Table 1. Here, we supplement three key differences against FCA below:
>
> (1) FCA uses frequency domain consistency constraint learning in spatial domain, and does not involve domain transformation and frequency calculation. We estimate the kernel entirely from the frequency domain. The advantages of frequency domain estimation enable us to estimate more complex and diverse types of kernels more accurately.
>
> (2) FCA can only deal with simple isotropic kernel, because the rough frequency consistency can only obtain blur intensity comparison but lack direction information. Our proposed method can process arbitrary kernel and estimate from a single test image, which makes our method more general and practical.
>
> (3) The core module of FCA is a frequency density comparator of siamese structure to obtain blur density consistancy. While our S2K's main body is an encode-decoder module based on conditional GAN, which is obviously different.
>
> Q4. Relevance of comparison between frequency and spatial domains.
>
> A4. Previously, it is the most common practice to estimate the blur kernel from the spatial domain for blind SR. Although some methods use frequency domain analysis as loss or constraint, they are all in the spatial framework. As the first attempt to perform kernel estimation completely in the frequency domain, we theoretically deduce that estimation in the frequency domain is more conducive to kernel estimation than that in the spatial domain (L135-L165). The visual comparison of spatial domain and frequency domain in Figure 1 and the experimental results in Table 2 and Figure 6 verify our superiority in dealing with this problem in the frequency domain.
> When demonstrating the advantages (L142-L143), our assumption is that the kernel is sparse but with no requirement for the form of the kernel.
>
> Q5. Writing issues.
>
> A5. We will carefully revise our work to improve writing and make it more readable in revision.
>
> Q6. Limitations and societal impact (see no limitation).
>
> A6. We actually already discussed the limitation of our method in Sec. 3  of our supplementary material (please refer to L13-L22 for details). For example, for non-uniform blur from sequential multi-frame mixing, our method, as well as other previous SOTA blind SR methods (FCA, FKP, IKC, DAN, etc.), may be not applicable. Last but not least, our discussion on limitation is well accepted by all of the other three reviewers, and we will include the discussion on limitation in our final version as suggested by the reviewers.

---

> > ### Comment · Reviewer_wMvH · 2021-09-07
> > **I raised my score**
> >
> > Dear authors,
> >
> > I found your answers very convincing, and based also on the reviews of my colleagues and your answers to their concerns, I decided to raise my score to 6. However, I still think that the writing is poor and, I case of being accepted, a careful revision of the the manuscript must be carried on. As for the degradation model, despite the fact that this model was used in other works, I insist that it is not a realistic model. As I said before, the way LR images are produced from HR images is by first convolving with the HR blur kernel, and then subsampling. Not the other way round.

---

> > > ### Author Response · Authors · 2021-09-11
> > > **Response to Reviewer wMvH.**
> > >
> > > Dear Reviewer,
> > >
> > > We sincerely thank you for your high appreciation on our response.
> > >
> > > Sure, during the past several weeks, we have already dedicated ourselves to improving the quality of our paper, such as correcting the typos and revising the manuscript regarding the insightful comments from all reviewers. For the degradation model, we will also include the comparison with the version that adopts convolution first and then down-sampling in our final version.

---

### Decision · Program_Chairs · 2021-09-27

**Decision:**

Accept (Poster)

**Comment:**

This paper tackles the problem of blind super-resolution when the images degraded by arbitrary blur kernels (assumed to be sparse both in frequency and space domains). The method proposes a new blur kernel estimation technique in the frequency domain as an end-to-end deep learning approach. The paper provides theoretical and experimental evidence to back the idea.

The core idea of the work is predicting degradation kernels in the frequency domain using deep neural networks. The proposed method is simple and effective, this was highlighted by all reviewers. The empirical evaluation seems convincing (and were further improved during the response period). The manuscript includes thorough ablation studies which clarify the benefits of each of the proposed components. This was highlighted by Reviewers wMvH and bKSR.

The authors provided a high quality rebuttal including new experiments and clarifying several points raised by the reviewers. In particular, the authors incorporated experiments on (i) mixed blur kernels, (ii) comparison to a more recent blind SR baselines, and (iii) quantitative and qualitative results on real historic images. All reviewers and the AC appreciate the authors response.

After the authors’ response, Reviewer wMvH increased the score to 6. The clarifications provided by the authors were considered sufficient. However, she/he expressed concerns regarding the papers presentation. Similarly Reviewer bKSR increased the score to 7. In the original review, the main concern was clarity.

he AC appreciates the authors response and counts that the authors will incorporate the the clarifications provided in their response. All reviewers provide an accepting score. The AC agrees and recommends accepting the paper.